# Analysis and Evaluation of Influencing Factors on Uniform Sowing of Wheat with Wide Seed Belt after Sowing and Soil Throwing Device

**Bokai Wang [1,2], Fengwei Gu [1,2,*], Zhichao Hu [3,*], Feng Wu [1,2], Xulei Chen [1,2] and Weiwen Luo [1,2]**

[1]  Nanjing Institute of Agricultural Mechanization, Ministry of Agriculture and Rural Affairs, Nanjing 210014, China
[2]  Key Laboratory of Modern Agricultural Equipment, Ministry of Agriculture and Rural Affairs, Nanjing 210014, China
[3]  Graduate School of Chinese Academy of Agricultural Sciences, Beijing 100083, China
*  Correspondence: gufengwei@caas.cn (F.G.); huzhichao@caas.cn (Z.H.)

**Abstract:** The uneven sowing of wheat on ground covered with rice straw in the rice–wheat rotation area in the middle and lower reaches of the Yangtze River has become a serious problem. Therefore, a test bed for throwing soil after sowing with a wide wheat seed belt was designed, which could complete the functions of straw crushing, straw lateral concentration and uniform sowing at one time. The discrete element simulation model of throwing soil after sowing with a wide wheat seed belt was established with rotary blade shaft speed, soil guide plate angle and soil retaining plate angle as variables. Taking the variation coefficient of wheat sowing depth and variation coefficient of sowing lateral uniformity as evaluation indexes, the effects of three variables on sowing uniformity were analyzed by single factor test and Box–Behnken test. The results of single factor test observed that when the rotating speed of rotary blade shaft was 260–300 rpm, the angle of soil guide plate was 36°–48°, the angle of soil retaining plate was 58°–74° and the experiment of utilizing a soil throwing and covering device with a wide seed belt after sowing revealed a good consistency of sowing depth and lateral uniformity effect. The Box–Behnken simulation experiment showed that the primary and secondary factors affecting the variation coefficient of wheat sowing lateral uniformity were the angle of soil guide plate, the rotation speed of rotary blade shaft, the angle of soil retaining plate and the angle of soil guide plate. When the rotation speed of rotary blade shaft, the angle of soil guide plate and the angle of soil retaining plate were 282.1 rpm, 42.4° and 65.5°, respectively, the soil throwing and covering device after sowing has the best seed-homogenizing effect. At this time, the variation coefficients of sowing depth and lateral uniformity in simulation test and field verification test were 4.35% and 4.57%, respectively, and 12.46% and 12.73%, respectively. The results of field verification test were basically consistent with those of the simulation test, which proved that the results of applying discrete element methods to optimize the soil-throwing device after sowing with a wide seed belt were credible. This study could provide a theoretical reference for the structure optimization of a soil-throwing device after sowing with a wide seed belt.

**Keywords:** wide seed belt; variation coefficient of sowing depth; sowing uniformity; rotary blade shaft speed; soil guide plate angle; soil retaining plate angle

## 1. Introduction

The wide-seed-belt sowing of wheat is a new technology in China. Compared with traditional mechanized sowing methods, it has great advantages, and a large number of scientific experiments had also proven this: Zhang et al. [1] thought that sowing with wide seed belt could improve the nitrogen absorption efficiency and tiller number of population in each growth period, and Zheng et al. [2] proved that the wide-seed-belt sowing of wheat not only saved time but was also low cost, and its yield was obviously higher than that of

furrow drilling. Zheng et al. [3] thought that sowing with wide seed belt was beneficial to reducing the phenomenon of straw entanglement and blocking, and it was one of the best choices in the mode of no-tillage and less tillage. However, by tracking the sowing experiments of wheat using a wide seed belt in the middle and lower reaches of the Yangtze River in China from 2018 to 2021, our research team found that there were problems of unstable sowing depth and uneven sowing in the sowing process of wheat with a wide seed belt, which had negative effects on the rooting and growth of wheat, plant height and yield and created a lot of production problem for wheat growers [4].

In recent years, international scholars have carried out a lot of research on the problems of unstable sowing depth and uneven sowing regarding the wide band sowing of wheat [5,6]. Zhao et al. [7] designed and tested the ditching depth control system of seeder, established the mathematical model of ditching depth, realized accurate ditching and real-time control and attained the synchronous copying of sowing depth. The indoor test results showed that the stability coefficients of ditching depth are 90.66% and 91.33%, respectively, under a sowing depth of 0.03 m and 0.05 m. Qin et al. [8] optimized the structural parameters, which effectively improved the sowing uniformity. Li et al. [9] used the high-speed stubble-cleaning mechanism of a double-shaft rotary tillage stubble-cleaning seeder; the wheat uniform sowing experiment was carried out in the field with large amount of rice straw and the effect was good. In order to solve the problem of the poor seed-metering uniformity of a precision seed-metering device for wide seed belt-planted wheat, Liu et al. [10] designed a hook-and-socket wheel type precision seed-metering device for wide seed belt-planted wheat, which was designed in combination with precision seed-metering technology, so that the seeds were evenly distributed in rows and not scattered among rows; In order to solve the problem of poor seeding uniformity of precision seeding devices for wheat planted using a wide seed belt, Niu et al. [11] designed a wheat seeder with a straw covering; by optimizing the rotary tillage mechanism, the degree of soil crushing was effectively improved, soil particles and wheat were evenly mixed and the seeding uniformity was effectively improved.

It can be seen that international scholars' research on the unstable and uneven sowing depth in the process of wide-seed-belt sowing mainly focused on the improvement of sowing structure, the optimization of seed metering device and the anti-blocking of the sowing parts, while there was little research on the uniform mixing mechanism of wheat–soil particles in the process of wide seed belt sowing. Moreover, the existing wheat seeder lacked a sowing depth control mechanism and uniform sowing mechanism, and the sowing depth could not be guaranteed by the lifting of the tractor's hydraulic suspension only. Some no-till seeders adopted post-profiling, which leaded to the phenomenon of profiling lag. There were some problems of unstable sowing depth and the uneven sowing lateral direction of wheat, which restricted the popularization and application of wide-seed-belt sowing in rice–wheat rotation areas in the middle and lower reaches of the Yangtze River in China.

In order to study the consistency and uniformity of sowing depth, our research team, based on the experimental data of wheat wide-seed-belt sowing in the middle and lower reaches of the Yangtze River in China, designed a soil-throwing test bed for wheat wide-seed-belt sowing to study the unstable sowing depth and the uneven lateral sowing of wheat in wide seed belt. The objectives of the study are: Firstly, this paper introduces the sowing mode of wheat wide seed belt in rice straw mulching field and the structure of the corresponding test bed, and analyzes its operation mechanism. The second research task is to carry out a single factor simulation design test and a Box–Behnken simulation test on the core system utilized on the test bed—the device which throws soil after sowing wheat with a wide seed belt—in order to obtain the best range and the best combination of various influencing parameters, achieve better sowing depth uniformity and uniform distribution effect of wheat and provide a theoretical basis for a field comparison test. The third research task is to verify the simulation test results through the field comparison

test of the soil-throwing device after wheat wide-seed-band sowing and also to provide theoretical support for the mechanized sowing of wheat in rice straw-covered fields.

## 2. Materials and Methods

### *2.1. Sowing Mode of Wheat Wide Seed Belt in Rice Straw Covered Land*

There are various agronomic models for sowing wheat using wide seed belts in China. Generally, the sowing width is 0.06–0.12 m and the spacing is 0.7–0.17 m. After years of field trials, our research team put forward a uniform sowing mode (Figure 1), in which the traditional sowing width is increased to 0.25 m and the spacing is set at 0.25 m, and the sowing mode was carried out in the rice–wheat rotation area in the middle and lower reaches of the Yangtze River.

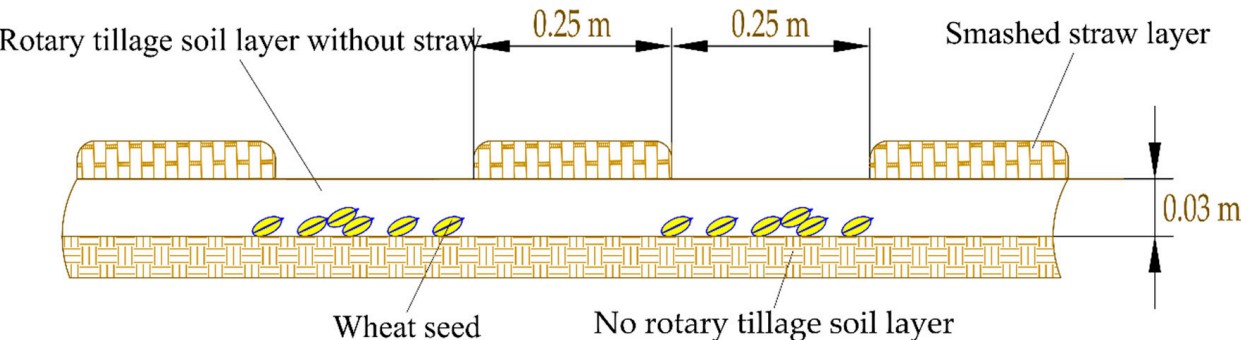

**Figure 1.** Sowing mode of wheat with wide planting belt in a rice straw mulching field.

The conventional mechanized sowing of wheat must include a straw crushing and returning machine, a rotary tiller and wheat seeder and various machines and tools to work the fields several times to complete the operation procedures of straw crushing, soil rotary tillage, fertilization and sowing, etc., which has a high production cost and is time-consuming. However, the sowing method of wheat with a wide seed belt covered with rice straw can complete the sowing in one instance, and it has the functions of saving agricultural time, reducing costs and increasing efficiency. The sowing carried out in this mode is conducive to the growing of larger wheat. The utilization rate of light energy is improved. Therefore, wheat has many secondary roots, thick seedlings and high spike rate, especially in the later stage, with a large green leaf area, long functional period, many spikes, big and full grains and a high 1000-grain weight, which has an obvious yield-increasing effect [12,13].

### *2.2. Overall Structure and Working Principle*

#### 2.2.1. Overall Structure

Figure 2 shows the overall structure of the test bed for throwing soil after sowing wheat with a wide seed belt in a rice straw-covered field. The mechanical system consists of a side-laying device for crushing straw into strips and a soil-throwing device after sowing. A fertilizer box was fixed above the laying device for crushing straw into strips, and a seed box was fixed above the soil throwing device after sowing with the wide seed belt. The mechanical system is connected to the tractor power through the first three suspension mechanisms, and the supporting power is 70–80 kW. There are 5 laying devices for crushing straw into strips and throwing soil after sowing. Each throwing soil device after sowing can sow 6 rows of wheat, and the number of rows of sowing is 6. The width of the test bed is 2.1 m, the rotary blade excircle diameter is 0.57 m, the size of the whole machine (length × width × height) is 2.2 m × 1.9 m × 1.3 m and the weight is 850 kg.

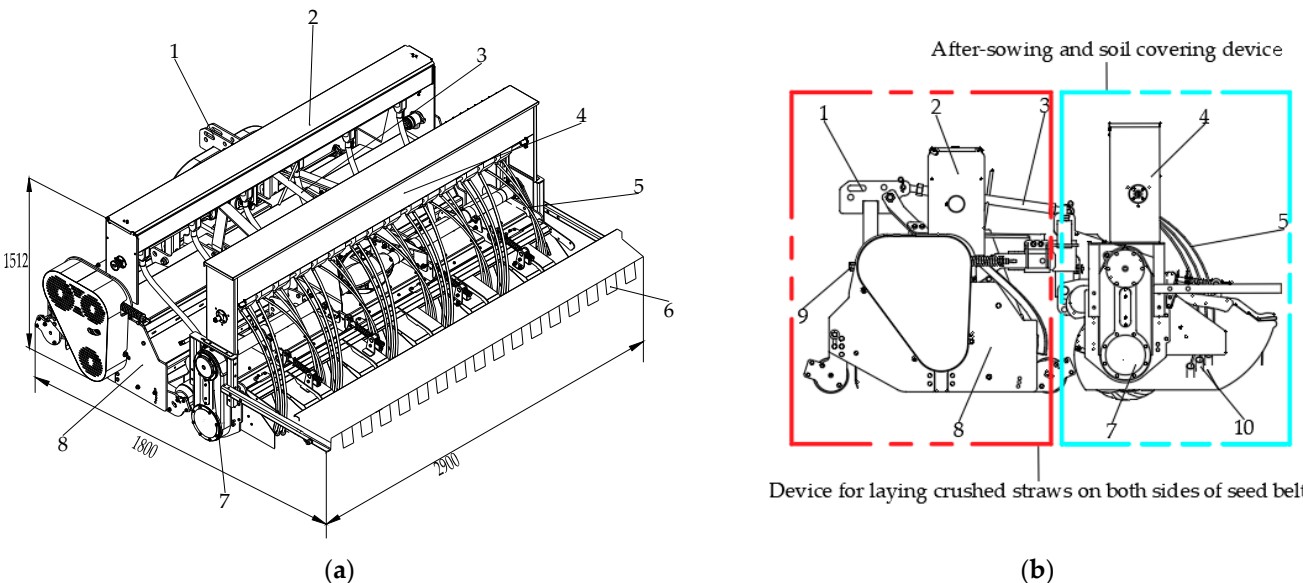

**Figure 2.** Overall structure of the wheat wide-seed-belt sowing and covering soil test bed: (1) hanging at the first three points; (2) fertilizer box; (3) hanging at the back three points; (4) seed box; (5) seed tube; (6) soil curtain; (7) transmission mechanism; (8) frame; (9) power input shaft; (10) seed guide tube. (**a**) Three-dimensional diagram; (**b**) side view.

### 2.2.2. Working Principle

Figure 3 shows the working principle of the wheat wide-seed-belt sowing and covering test bed. When the test bed moves forward, the tractor drives over the test bed to move forward, and the straw-crushing mechanism rotates reversely to crush all the straws in the working width. The rear strip laying device moves forward and slides in the crushed straws, dividing and gathering the flowing broken straws to both sides to form longitudinal straw belts on both sides. A straw-free wide seed belt with a width of 0.25 m is formed between the straws (below the strip-laying device), and the fertilizer is transported into the wide seed belt through the fertilizer pipes. After the wide seed belt is sown, the soil throwing device finishes rotary tillage in the straw-free wide seed belt. Meanwhile, the wheat seeds are evenly sown on the soft soil after rotary tillage, and the soil thrown after rotary tillage evenly covers the wheat seeds after rotary landing.

The working environment of the test bed is a field covered with a full amount of straw. It innovatively integrates the functions of laying straw in strips after crushing, spreading wheat evenly with a wide seed belt, and covering soil evenly after sowing. It scientifically realizes that in the sowing belt formed by laying straw in strips, the soil is rotary tilled and then thrown across the seed guide tube, and the seeds are uniformly sown on the soft seed bed treated by rotary tilling, and the seeds are covered with flying soil, thus innovatively attaining wide-seed-belt sowing.

### 2.3. *Working Principle of Soil Throwing and Covering Device after Sowing*

Figures 4 and 5 shows the working principle diagram of the soil-throwing and covering device for wheat wide-seed-belt after sowing, which is mainly composed of rotary tillage mechanism, seed metering pipe, upper soil retaining plate, side baffle plate, rear soil retaining plate, seed dropping limit pipe and soil guide plate. The rotary tillage width of rotary tillage mechanism in soil throwing and covering device after sowing wide seed belt is 0.25 m, which is the same as that of a straw-free wide seed belt formed by a straw crushing and laying device. The two side baffles are welded and fixed on both sides of rotary tillage mechanism, the upper part of which is equipped with a floating adjustable upper cover plate and the rear part of which is welded and fixed with a rear retaining plate, so that the fine soil thrown by rotary tillage can move in a closed space. The left

and right side baffles are welded and fixed with soil guide plates, and the lower parts are, respectively, fixed with three seed dropping limit pipes; six seed dropping limit pipes are evenly distributed in the wide seed belt; and six seed dropping pipes are inserted into the seed dropping limit pipes at a one-to-one ratio.

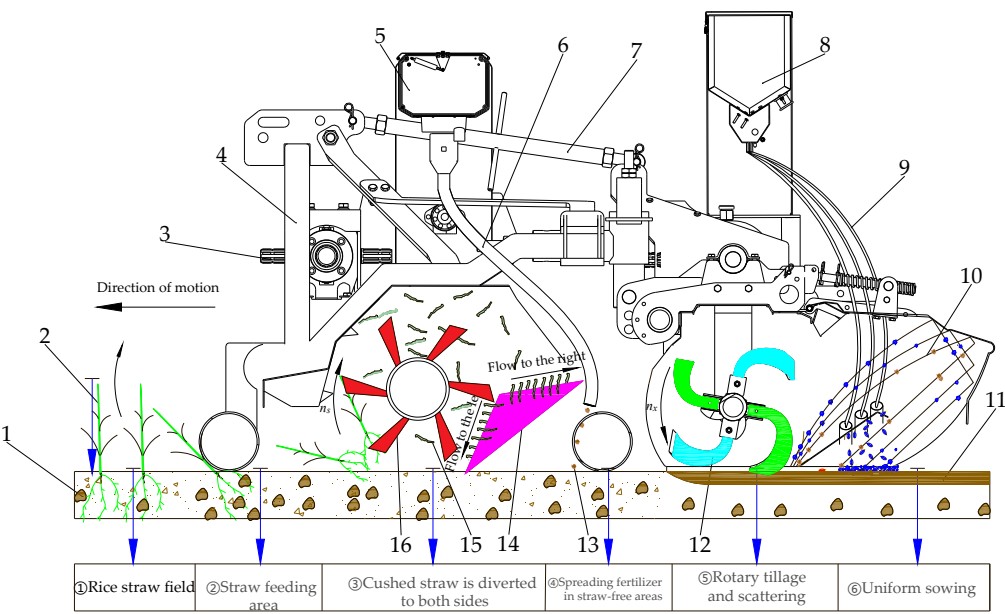

**Figure 3.** Working principle of wheat wide-seed-belt sowing and covering the soil test bed: (1) rough soil in non-rotary tillage area; (2) rice straw; (3) power input shaft; (4) rack; (5) fertilizer box; (6) fertilizer tube; (7) three-point suspension at the back; (8) seed box; (9) seed tube; (10) soil; throwing track; (11) fine ground soil after rotary tillage; (12) rotary tillage mechanism; (13) wheat; (14) strip laying device; (15) broken straw; (16) straw-crushing mechanism.

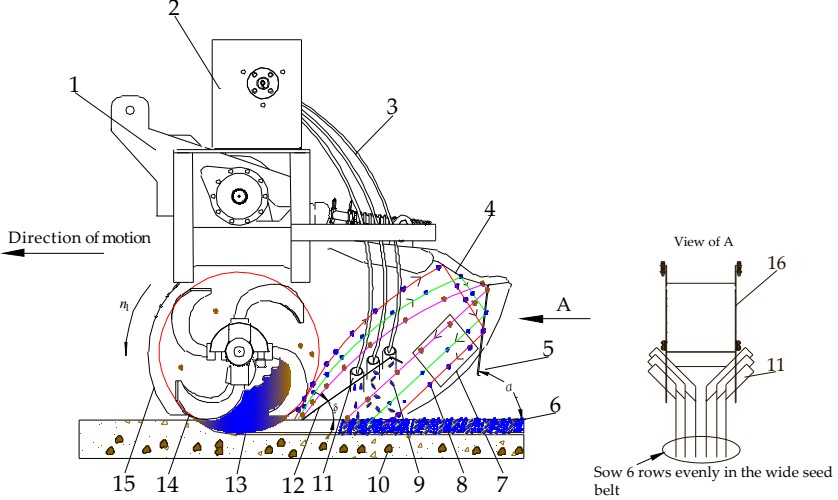

**Figure 4.** Working principle diagram of a soil-throwing and covering device after wheat wide seed belt: (1) rack; (2) seed box; (3) seed tube; (4) upper retaining plate (5) rear retaining plate; (6) finely ground soil after rotary tillage; (7) soil throwing track; (8) finely ground soil particles; (9) wheat; (10) rough soil in non-rotary tillage area; (11) seed guide tube (12) soil guide plate; (13) thrown finely ground soil; (14) rotary blade; (15) rotary blade baffle; (16) side baffle.

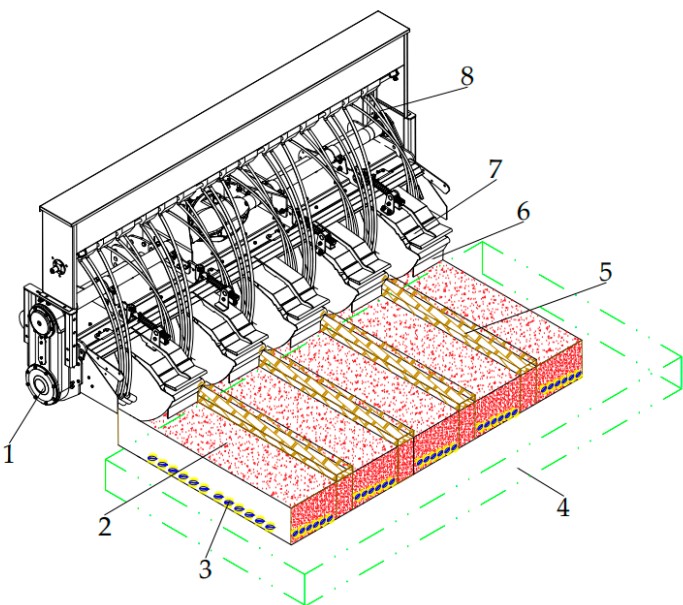

**Figure 5.** Axial schematic diagram: (1) gear train; (2) seed box; (3) seeds; (4) sowing space; (5) straw-crushing concentrated area; (6) soil-retaining plate; (7) upper baffle; (8) seed tube.

When the machine moves forward, the soil splashed by the rotary tiller mechanism is guided by the soil guide plate, which flies obliquely behind the seed guide tube. Under the soil guide plate, there is a finely ground soil area after rotary tillage. The wheat seeds enter the seed guide tube through the seed-metering tube and are directly and evenly sown with the wide seed belt for 6 rows. Finally, the thrown soil hits the upper cover plate and the rear retaining plate and forms soil "rain" along the rear retaining plate, which scatters and drops the wheat seeds from the wide seed belt evenly.

### 2.4. Simulation Test of Soil Throwing and Covering Device after Sowing

2.4.1. Design of Simulation Test

In order to analyze the soil-throwing performance of the rotary blade and verify the uniform covering effect of soil thrown by rotary tillage on the wheat after sowing, the discrete element method was used to simulate the soil-throwing performance of the soil-throwing device after sowing. A soil model of 0.2 m (length) × 0.3 m (width) × 0.1 m (height) was established by using EDEM 2020 simulation software, and the three-dimensional structure of the soil-throwing device after sowing was introduced into EDEM2020, and the simulation model of soil throwing device after sowing was obtained. Referring to the actual operation conditions, the soil in the experimental site was sandy loam, the soil moisture content was between 32.5% and 43.1% and the adhesion of soil to the soil throwing device after sowing using the wide seed belt was small. Therefore, Hertz–Mindlin (no slip) was selected as the contact model between the soil and the soil covering device (steel) after the wide seed belt was used for sowing [14], and the linear cohesion contact model, which gave the soil normal cohesion, was selected as the contact model. The cohesive energy density among soil particles was set at 6300 J/cm$^2$ [15], and the soil particles were modeled by spherical particles with a diameter of 0.8–1.2 mm. The rotary tillage depth was set at 0.05 m, the advance speed of rotary tillage and covering soil at 4 km/h, the simulation step size at $4.0 \times 10^{-6}$ s and the total simulation time at 15 s. The distribution of soil particles and wheat was generated by different particle factory. See Table 1 for the setting of basic structure parameters of each particle, and Table 2 for the setting of dynamic/static friction factors among various types of particles. Figure 6 is a simulation model of the soil-throwing and covering device after sowing.

**Table 1.** Basic structural parameters of each particle.

| Particle Type | Number of Small Balls | Draw Ratio | Particle Diameter Range |
|---|---|---|---|
| Wheat | Unlimited | 2.04 | 0.001–0.003 |
| Soil particles | Unlimited | 1.32 | 0.005–0.015 |

**Table 2.** Dynamic/static friction factors among various types of particles.

| Dynamic/Static Friction Factor | Wheat | Soil Particles | Mechanical Part |
|---|---|---|---|
| Wheat | 0.09/0.16 | | |
| Soil particles | 0.09/0.16 | 0.09/0.17 | |
| Mechanical part | 0.09/0.16 | 0.09/0.17 | 0.09/0.15 |

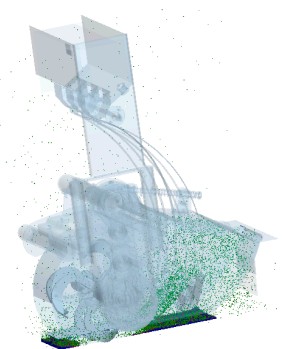

**Figure 6.** Simulation model of soil throwing and covering device after sowing.

2.4.2. Single Factor Simulation Test Design

According to the above mechanism analysis, the rotating speed of rotary blade shaft, the angle of soil guide plate and the angle of soil retaining plate had a decisive influence on the quantity and speed of soil particles thrown, so the influencing factors of single factor test were determined as the rotating speed of rotary blade shaft, the angle of soil guide plate and the angle of soil retaining plate, and, according to previous experimental experience, the rotating speeds of rotary blade shaft were set at 190 rpm, 210 rpm, 230 rpm, 250 rpm, 270 rpm, 290 rpm, 310 rpm and 330 rpm. The angle levels of the soil guide plate were set at 18°, 24°, 30°, 36°, 42°, 48°, 54° and 60°, and the angle levels of the soil retaining plate were set at 40°, 45°, 50°, 55°, 60°, 65°, 70°, 75° and 80°.

2.4.3. Box–Behnken Simulation Test Design

After the single factor test established the range of three influencing factors, the simulation test set the same soil particle environment and advancing speed as the previous single factor test. The rotation speed $X_1$ of the rotary blade shaft, the angle $X_2$ of the soil guide plate and the angle $X_3$ of the soil-retaining plate were selected as experimental factors. Taking the variation coefficient of sowing depth $Y_1$ and the variation coefficient of sowing lateral uniformity $Y_2$ as indexes, the orthogonal experiment of three factors and three levels was carried out [16,17]. Table 3 displays the test factors and codes.

**Table 3.** Test Factors and Coding.

| Encode | Factor | | |
|---|---|---|---|
| | Rotary Blade Shaft Speed $X_1$/(rpm) | Angle of Soil Guide Plate $X_2$/(°) | Angle of Retaining Plate $X_3$/(°) |
| −1 | 260 | 36 | 58 |
| 0 | 280 | 42 | 66 |
| 1 | 300 | 48 | 74 |

According to the Box–Behnken test principle, the test scheme and results are shown in Table 4. The data were analyzed by quadratic polynomial regression with Design-expert 12.0 software, and the correlation and interaction effects of various factors were analyzed by response surface analysis.

**Table 4.** Experiment design and response values.

| No. | Codes | | | Response Values | |
| --- | --- | --- | --- | --- | --- |
| | Rotary Blade Shaft Speed $X_1$ | Angle of Soil Guide Plate $X_2$ | Angle of Retaining Plate $X_3$ | Variation Coefficient of Sowing Depth $Y_1$/% | Variation Coefficient of Lateral Uniformity of Sowing $Y_2$/% |
| 1 | 0 | 0 | 0 | 4.23 | 12.61 |
| 2 | 0 | −1 | −1 | 6.24 | 16.74 |
| 3 | 0 | 0 | 0 | 4.71 | 12.34 |
| 4 | 1 | 0 | −1 | 6.69 | 17.62 |
| 5 | −1 | 0 | −1 | 7.75 | 19.07 |
| 6 | −1 | 1 | 0 | 8.13 | 19.63 |
| 7 | 1 | 0 | 1 | 6.81 | 18.12 |
| 8 | 0 | −1 | 1 | 6.42 | 17.07 |
| 9 | 0 | 1 | −1 | 5.87 | 16.17 |
| 10 | −1 | −1 | 0 | 8.54 | 20.12 |
| 11 | 0 | 0 | 0 | 4.17 | 12.58 |
| 12 | 1 | 1 | 0 | 7.06 | 18.19 |
| 13 | −1 | 0 | 1 | 7.94 | 19.37 |
| 14 | 0 | 0 | 0 | 4.55 | 12.03 |
| 15 | 1 | −1 | 0 | 7.41 | 18.54 |
| 16 | 0 | 1 | 1 | 6.09 | 16.39 |
| 17 | 0 | 0 | 0 | 4.31 | 12.94 |

### 2.4.4. Detection Method of Variation Coefficient of Sowing Depth

Using the post-processing function (Selection) of EDEM 2020, three equally spaced grids (three layers) were set in the longitudinal direction (Y direction), and the distances from the ground were 0–0.012 m, 0.012–0.024 m and 0.024–0.036 m, respectively. The plane size of each grid was 252 mm × 300 mm, and then the statistical function in the software could be used, Figure 7 shows this method.

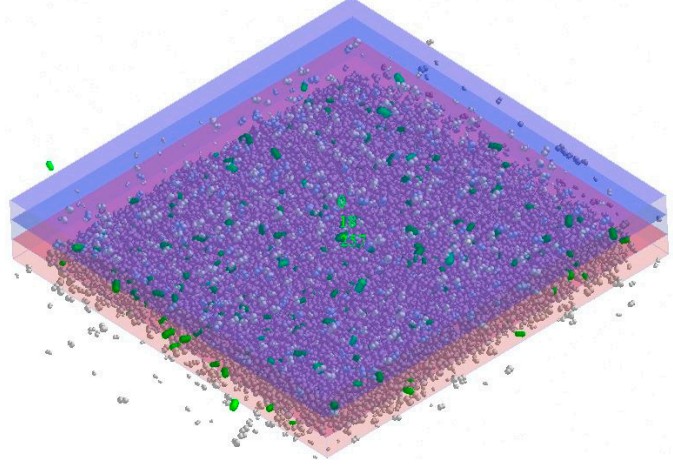

**Figure 7.** Three equally spaced grids (three layers) are set in the longitudinal direction (y direction).

Equation (1) was used to calculate the standard deviation of the longitudinal number of wheat:

$$\begin{cases} s_1 = \sqrt{\dfrac{\sum\limits_{k=1}^{3} (z_k - \overline{z})^2}{2}} \\ \overline{z} = \dfrac{1}{3} \sum\limits_{k=1}^{3} z_k \end{cases} \tag{1}$$

Here, $s_1$ is the standard deviation of the longitudinal wheat number; $\overline{z}$ is the average number of wheat in each layer and $Z_k$ is the number of layers. $Z_k$ is the number of wheat in the $k$ layer.

Equation (2) was used to calculate the variation coefficient of the wheat-sowing depth.

$$\varsigma = \frac{s_1}{\overline{z}} \times 100\% \tag{2}$$

Here, $\varsigma$ is the standard deviation of the longitudinal wheat number.

### 2.4.5. Testing Method of Lateral Uniformity of Sowing

Using the post-processing function (Selection) of EDEM 2020, six equally spaced grids were set in the longitudinal direction (perpendicular to the advancing direction of the machine), and the width $\times$ length of each grid was 0.360 m $\times$ 0.042 m After that, the amount of wheat in each grid could be counted by the statistical function in the software.

The Equation (3) was used to calculate the average number of wheat particle in the grid cell [18].

$$\begin{cases} \overline{x}_i = \dfrac{1}{5} \sum\limits_{j=1}^{5} x_{ij} \\ \overline{x} = \dfrac{1}{6} \sum\limits_{i=1}^{6} x_i \end{cases} \tag{3}$$

where, $\overline{x}_i$ is the average amount of wheat in all grid cells in row I; $x_{ij}$ is the amount of wheat in the grid in row $i$ and column $j$; $\overline{x}$ is the average amount of wheat in all rows; and $x_i$ is the amount of wheat in the grid in row $i$.

Equation (4) was used to calculate the standard deviation of wheat grain number.

$$s = \sqrt{\frac{\sum\limits_{i=1}^{n} (x_i - \overline{x})^2}{n-1}} \tag{4}$$

Then the calculation equation of variation coefficient of wheat sowing lateral uniformity is as follows:

$$\sigma = \frac{s}{\overline{x}} \times 100\% \tag{5}$$

Here, $\sigma$ is the variation coefficient of wheat sowing lateral uniformity.

### 2.5. Field Validation Test

To further test the variation coefficient of wheat-sowing depth and the variation coefficient of sowing lateral uniformity under the optimal combination conditions in the simulation test, a field verification test was carried out. The field verification test was conducted at the Xuzhou Academy of Agricultural Sciences in October 2020. The last crop in the experimental field was rice, and the straw was returned to the field multiple times. After the treatment, the straw was finely divided, and the straw coverage was 3.8 kg/m$^2$. Before the test, the water content of 0–0.1 m, 0.1–0.2 m and 0.2–0.3 m soil was 13.78%, 19.25% and 22.85%, respectively, and the soil firmness was 1.31, 1.64 and 2.7 MPa, respectively.

The measurement of the variation coefficient of wheat sowing depth: because the thickness of covering soil determines the sowing depth of seeds, the vertical distance of each seed from the ground was measured as the sowing depth. The measurement method of the

variation coefficient of sowing lateral uniformity was similar to that used in the simulation test. Ten areas with a diameter of 0.252 m × 0.300 m were randomly selected in the wide seed belt along the direction of the vertical machine advance, and six equally spaced grids were set in each area. Then, the variation coefficient of sowing lateral uniformity was calculated by Equation (6). The tractor used was a John Deere 1204 tractor, and the working speed was 1.5 m/s. According to the climatic conditions and sowing time in the middle and lower reaches of the Yangtze River, "Nanjing 45" a rice straw mulching field was selected for wheat operation. According to the method specified in the NY/T 1768-2009 Technical Specification for the Quality Evaluation of No-Tillage Seeders and the GB/T24675.6-2009 straw-crushing and -returning machine for conservation tillage machinery, the two indexes of variation coefficient of wheat-sowing depth and the variation coefficient of sowing lateral uniformity were verified [19,20] Testing equipment included electronic scale, soil hardness tester, vernier caliper, stopwatch, tape measure and shovel, etc.

## 3. Results

### 3.1. Effect Analysis of Rotary Blade Shaft Throwing Soil and Covering Seeds

Figure 8 shows that six rows of wheat seeds can be well planted at a relatively consistent depth, and this effect is beneficial to keep a consistent growth space, It can be seen from Figure 9b that the lateral distribution position of wheat seeds is basically on or near the ideal position line, which reflects and illustrates the lateral distribution effect of the device well.

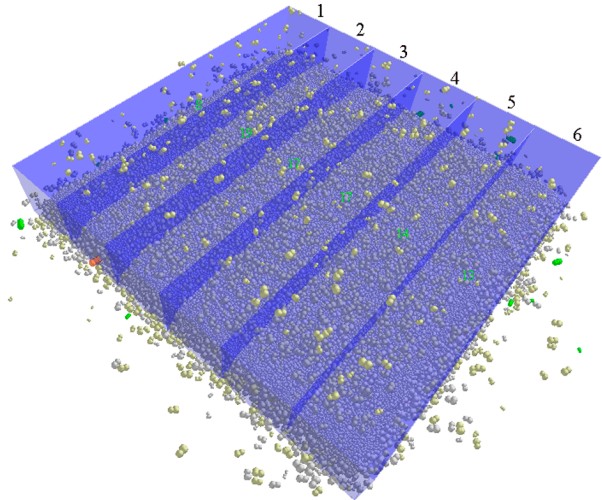

**Figure 8.** Six equally spaced grids in the longitudinal direction (perpendicular to the advancing direction of the machine) are set up.

### 3.2. Single Factor Result Analysis

3.2.1. Single Factor Results of Variation Coefficient of Sowing Depth

Figure 10a shows the variation trend of the variation coefficient of sowing depth at different rotating speeds of the rotary blade shaft. With the rotating speed of rotary blade shaft gradually increasing, the variation coefficient of the sowing depth first gradually decreases and then increases. When the rotating speed of rotary blade shaft is 260–300 rpm, the variation coefficient of sowing depth is small. Figure 10b shows the variation trend of the variation coefficient of sowing depth at the different angles of the soil guide plate. With the increase in the angle of the soil guide plate, the variation coefficient of sowing depth first decreases and then increases. When the angle of soil guide plate is 36°–48°, the variation coefficient of sowing depth is smaller. Figure 10c shows the variation trend of the variation coefficient of the sowing depth at the different angles of the retaining wall. With the increase in the retaining wall angle, the variation coefficient of sowing depth

first decreases and then increases. When the retaining wall angle is 58°–74°, the variation coefficient of the sowing depth is smaller.

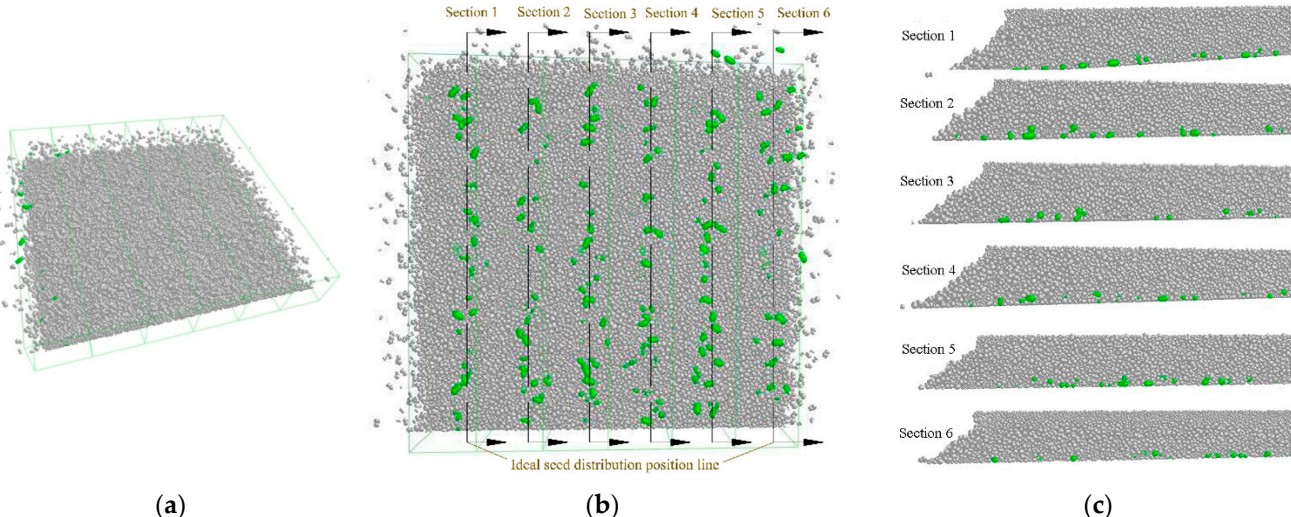

**Figure 9.** Effect analysis of throwing soil and covering seeds. (**a**) Top view when t = 0.1 s; (**b**) bottom view when t = 0.1 s; (**c**) t6 sectional views when t = 0.1 s.

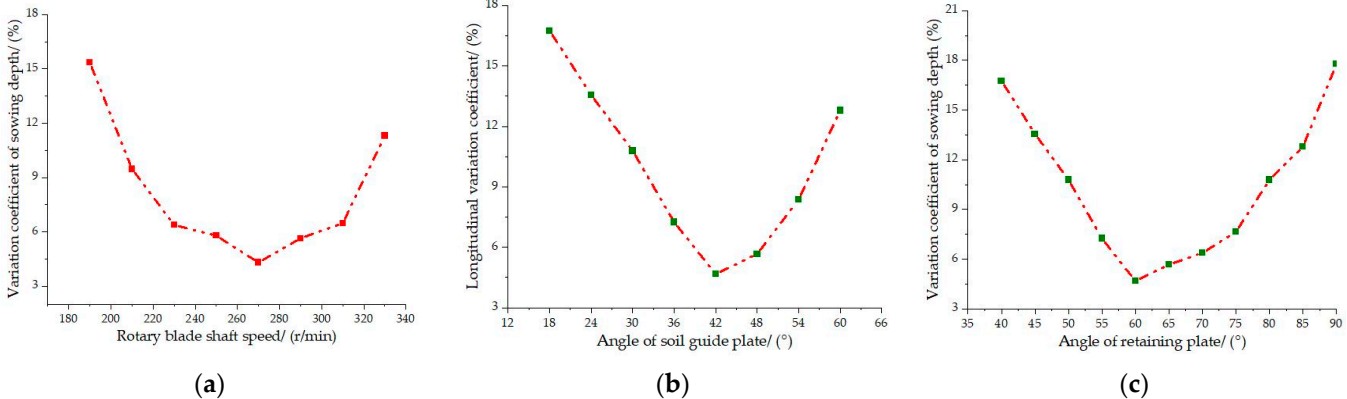

**Figure 10.** Single factor test results of the variation coefficient of sowing depth: (**a**) the relationship between the rotation speed of the rotary blade shaft and the variation coefficient of sowing depth; (**b**) the relationship between the angle of soil guide plate and the variation coefficient of sowing depth; (**c**) the relationship between the angle of the soil-retaining plate and the variation coefficient of the sowing depth.

3.2.2. Single Factor Results of Lateral Uniformity

Figure 11a shows the variation trend of the variation coefficient of the seed sowing lateral uniformity at different rotational speeds. With the rotational speed of rotary blade increases gradually, the variation coefficient of seed sowing lateral uniformity decreases at first and then increases gradually. When the rotational speed of rotary blade is 270 rpm, the variation coefficient of seed sowing lateral uniformity is the smallest, indicating that the wide seed belt sowing effect is better when the rotational speed is around 270 rpm. The average number of seeds in the grid of different rows and single columns in Figure 11d also reflects this trend. When the rotating speed of the rotary blade shaft is less than 270 rpm or greater than 270 rpm, the number of wheat seeds is more on both sides and less in the middle. The reason for this may be that the rotary blade has the strongest soil-throwing ability in the middle.

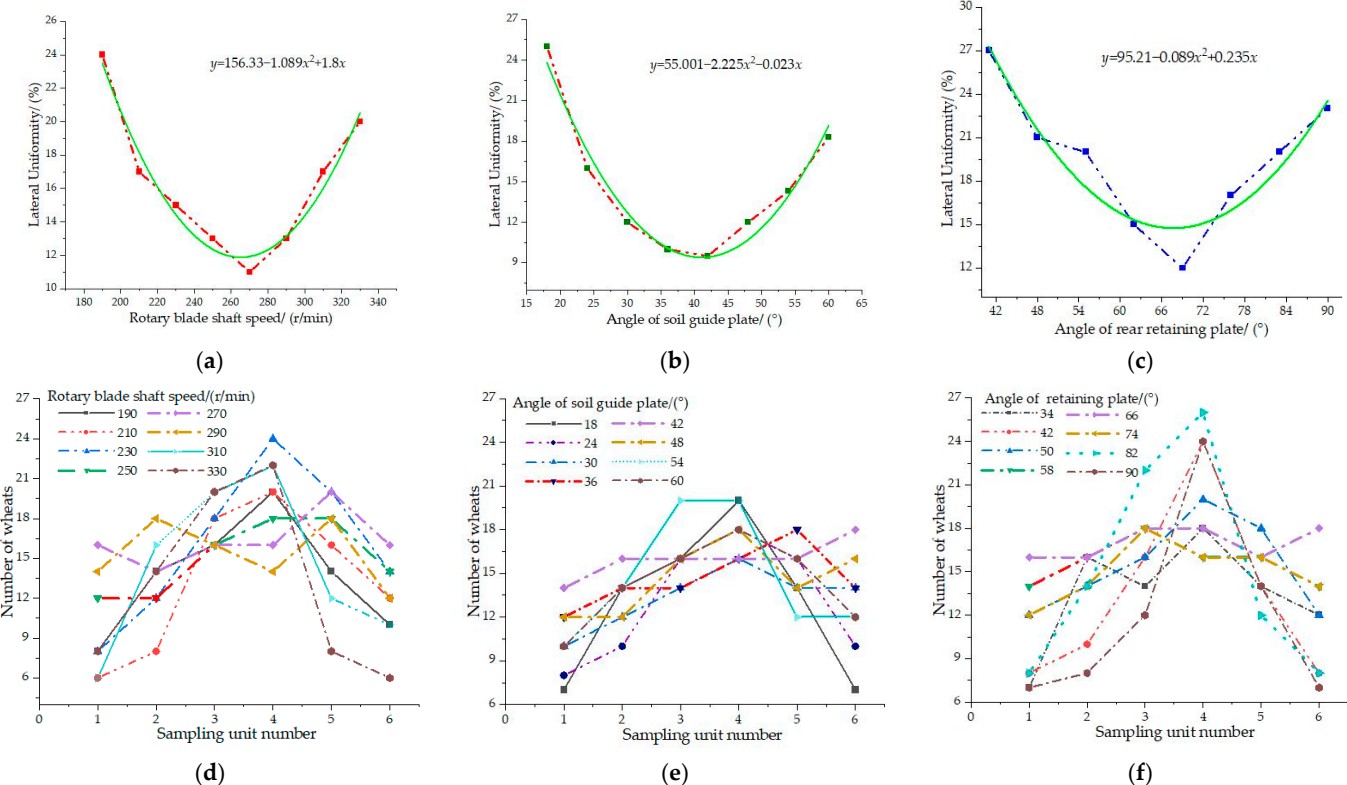

**Figure 11.** Single factor results of lateral uniformity. (**a**) The relationship between the rotation speed of rotary blade shaft and the lateral uniformity of wheat; (**b**) the relationship between the angle of the soil guide plate and the lateral uniformity of wheat; (**c**) the relationship between the angle of the soil-retaining plate and the lateral uniformity of wheat; (**d**) the relationship between the rotating speed of the rotary blade shaft and the number of seeds in each unit; (**e**) the relationship between angle of the soil guide plate and number of seeds in each unit; (**f**) the relationship between angle of the soil-retaining plate and number of seeds in each unit.

Figure 11b shows the variation trend of the variation coefficient of the lateral uniformity of seed sowing at different angles of soil guide plate. With the increasing of the angle of the soil guide plate, the variation coefficient of the lateral uniformity of seed first decreases and then increases gradually. When the angle of soil guide plate is 42°, the variation coefficient of the lateral uniformity of seed sowing is the smallest, indicating that the wide seed belt sowing effect is better when the angle of the soil guide plate is 42°. The average amount of wheat in different rows and single columns in Figure 11e also reflects this trend. When the angle of the soil guide plate is less than 42° or greater than 42°, the number of wheat grains is less on both sides and more in the middle. The reason may be that too large or too small an angle of the soil guide plate will weaken the flying distance of soil particles in the air and the whirling and scattering effect after impact.

Figure 11c shows the variation trend of the variation coefficient of the lateral uniformity of seed sowing at the different angles of the soil guide plate. With the increasing of the angle of the soil guide plate, the variation coefficient of the lateral uniformity of seed firstly decreases and then increases gradually. When the angle of soil retaining plate is 66°, the variation coefficient of the lateral uniformity of seed sowing is the smallest, indicating that the sowing effect of the wide seed belt is better when the angle of soil retaining plate is 66°. The average number of wheat grains in the grid of different rows and single columns in Figure 11f also reflects this trend. When the angle of the retaining wall is less than 66° and greater than 42°, the number of wheat grains is less on both sides and more in the middle. The reason for this may be that the angle of the retaining wall is too large or too small, which will weaken the rebound effect of soil particles after impact, and then affect

its scattering effect. Finally, it will affect the effect of soil covering wheat and increase the lateral uniformity of wheat grains.

*3.3. Single Factor Result Analysis*

3.3.1. Establishment and Test of Regression Model

Using Design-expert 12.0 software to carry out multiple regression fitting analysis on the data in Table 5, the response surface regression model of $Y_1$, $Y_2$ to $X_1$, $X_2$ and $X_3$ is established, as shown in Equations (6) and (7), and the variance analysis of the regression equation is carried out as shown in Table 5.

$$Y_1 = 4.39 - 0.55X_1 - 0.18X_2 + 0.09X_3 + 0.02X_1X_2 \\ -0.02X_1X_3 + 0.01X_2X_3 + 2.27X_1{}^2 + 1.12X_2{}^2 + 0.64X_3{}^2 \tag{6}$$

$$Y_2 = 12.5 - 0.72X_1 - 0.26X_2 + 0.17X_3 + 0.04X_1X_2 \\ +0.05X_1X_3 - 0.03X_2X_3 + 4.29X_1{}^2 + 2.33X_2{}^2 + 1.76X_3{}^2 \tag{7}$$

**Table 5.** Variance analysis of regression equation.

| Variance Source | Variation Coefficient of Sowing Depth $Y_1$ | | | | Coefficient of Variation of Lateral Uniformity $Y_2$ | | | |
|---|---|---|---|---|---|---|---|---|
| | Sum of Squares | Freedom | *F* | *P* | Sum of Squares | Freedom | *F* | *P* |
| model | 33.79 | 9 | 125.26 | <0.0001 | 129.05 | 9 | 200.65 | <0.0001 |
| $X_1$ | 2.41 | 1 | 80.38 | <0.0001 | 4.09 | 1 | 57.23 | 0.0001 |
| $X_2$ | 0.2664 | 1 | 8.89 | 0.0205 | 0.5460 | 1 | 7.64 | 0.0279 |
| $X_3$ | 0.0630 | 1 | 2.10 | 0.1903 | 0.2278 | 1 | 3.19 | 0.1174 |
| $X_1X_2$ | 0.0009 | 1 | 0.0300 | 0.8673 | 0.0049 | 1 | 0.0686 | 0.8010 |
| $X_1X_3$ | 0.0012 | 1 | 0.0409 | 0.8455 | 0.0100 | 1 | 0.1399 | 0.7194 |
| $X_2X_3$ | 0.0004 | 1 | 0.0133 | 0.9113 | 0.0030 | 1 | 0.23 | 0.8428 |
| $X_1{}^2$ | 21.63 | 1 | 721.85 | <0.0001 | 77.36 | 1 | 1082.49 | <0.0001 |
| $X_2{}^2$ | 5.32 | 1 | 177.57 | <0.0001 | 22.93 | 1 | 320.91 | <0.0001 |
| $X_3{}^2$ | 1.71 | 1 | 56.96 | 0.0001 | 13.02 | 1 | 182.25 | <0.0001 |
| Residual | 0.2098 | 7 | | | 0.5002 | 7 | | |
| Lack of fit | 0.0015 | 3 | 0.0094 | 0.9985 | 0.0416 | 3 | 0.1210 | 0.9430 |
| Pure error | 0.2083 | 4 | | | 0.4586 | 4 | | |
| Cor total | 34.00 | 16 | | | 129.55 | 16 | | |

Here, $Y_1$ is the variation coefficient of sowing depth, %, and $Y_2$ is the variation coefficient of lateral uniformity, %.

It can be seen from Table 5 that the *P* scores of the variation coefficient of sowing depth and the variation coefficient of lateral uniformity are all less than 0.05, which indicates that the two models have a significant influence. The determination coefficient $R^2$ values are 0.9938 and 0.9961, respectively, which indicates that more than 99% of the response values can be explained by the two models, and the p values of the mismatching items are all greater than 0.05, so the mismatching is not significant. Therefore, the model can predict the working parameters of the seeding system. According to the change of the response surface map of the two models, it can be gathered that the primary and secondary order of the influence of each factor on the variation coefficient of sowing depth and lateral uniformity is $X_1$, $X_2$ and $X_3$, that is, the rotation speed of rotary blade shaft, the angle of soil guide plate and the angle of soil-retaining plate.

Figure 12a shows that the variation coefficient of sowing depth decreases first and then increases with the increase of rotary blade shaft speed and decreases first and then increases with the increase of soil guide plate angle, and the influence of rotary blade shaft speed on the variation coefficient of sowing depth is greater than that of soil guide plate angle. Figure 12b shows that the variation coefficient of sowing depth first decreases and then increases with the increase of rotary blade shaft speed, and the influence of

rotary blade shaft speed on the variation coefficient of sowing depth is greater than that of retaining plate angle. Figure 12c shows that the variation coefficient of the sowing depth first decreases and then increases with the increase of the angle of soil guide plate, the influence of the angle of the soil guide plate on the variation system of the sowing depth is greater than that of the angle of soil-retaining plate.

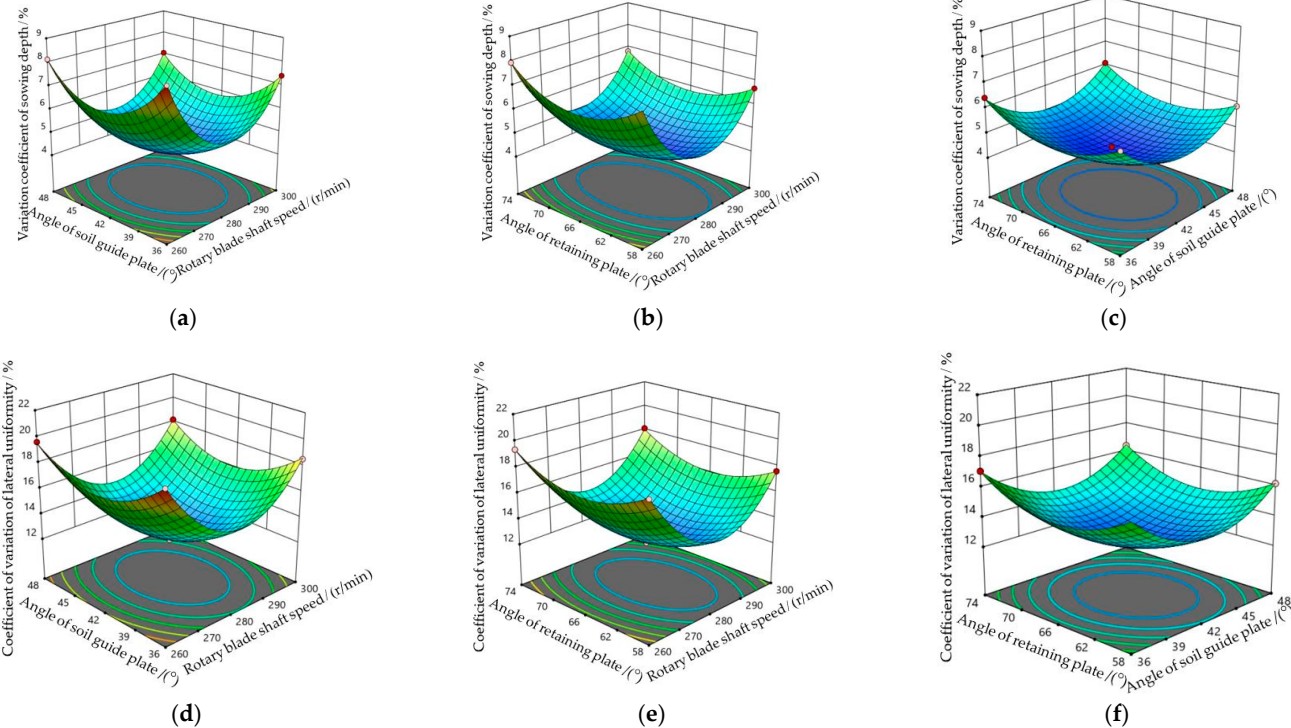

**Figure 12.** Response surface of interaction factors to the variation coefficient of sowing depth and variation coefficient of lateral uniformity. (**a**) Response of rotating blade shaft speed and angle of soil guide plate to sowing depth coefficient; (**b**) Response of angle of soil guide plate and angle of retaining plate to sowing depth coefficient; (**c**) Response of rotating blade shaft speed and angle of retaining plate to sowing depth coefficient; (**d**) Response of rotating blade shaft speed and angle of soil guide plate to lateral uniformity; (**e**) Response of angle of soil guide plate and angle of retaining plate to lateral uniformity; (**f**) Response of rotating blade shaft speed and angle of retaining plate to lateral uniformity.

Figure 12d shows that the variation coefficient of lateral uniformity first decreases and then increases with the increase of rotary blade shaft speed, and the variation coefficient of lateral uniformity first decreases and then increases with the increase of soil guide plate angle, and the influence of rotary blade shaft speed on the variation coefficient of lateral uniformity is greater than that of soil guide plate angle. Figure 12e shows that the variation coefficient of lateral uniformity decreases first and then increases with the increase of the rotary blade shaft speed, and the influence of rotary blade shaft speed on sowing depth variation system is greater than that of the retaining plate angle. Figure 12f shows that the variation coefficient of the lateral uniformity decreases first and then increases with the increase of the angle of soil guide plate, and the influence of the angle of the soil guide plate on the variation coefficient of sowing depth is greater than that of the angle of the soil retaining plate.

### 3.3.2. Parameter Optimization

According to the analysis of the above test results, in order to further improve the operational performance of the winnowing system, the minimum variation coefficient of sowing depth, $Y_1$, and the maximum variation coefficient of lateral uniformity, $Y_2$,

are taken as optimization indexes, and a full-factor quadratic regression Equation (8) of performance indexes is established to optimize the target and determine the optimal operational parameters.

$$minG(X) = \begin{cases} Y_1(X_1, X_2, X_3) \\ Y_2(X_1, X_2, X_3) \end{cases} \tag{8}$$

among

$$\begin{cases} 260 \leq X_1 \leq 300 \\ 36 \leq X_2 \leq 48 \\ 58 \leq X_3 \leq 74 \end{cases}$$

The optimal parameter combination of the minimum sowing depth variation coefficient, $Y_1$, and the lateral uniformity variation coefficient, $Y_2$, which meets the constraint conditions can be obtained by using the optimization solution module of Design-Expert 12.0. The optimal parameter combination is: rotary blade shaft speed 282.1 rpm, soil guide plate angle 42.4° and soil retaining plate angle 65.5°, and the corresponding variation coefficients of sowing depth and lateral uniformity are 4.35% and 12.46%, respectively.

### 3.4. Analysis of Field Test Results

As can be seen from Table 6, after the field verification test, three seed belts were randomly selected, and four points were randomly selected from each seed belt for testing. The variation coefficients of sowing depth of three seed belts are 4.93%, 4.61% and 4.17% respectively, while the variation coefficients of lateral uniformity are 12.52%, 13.48% and 12.19% respectively. The variation coefficient of sowing depth (4.57%) and the variation coefficient of lateral uniformity (12.73%) were higher than those of the simulation optimization results. Compared with simulation, the volume of soil after machine operation was relatively uneven, and the qualified rate and stability of sowing depth were reduced after covering seeds. Before the machine operation in the field, the flatness of the ground surface was lower than that of the soil surface in the simulation model. In the field test, with the change of depth and topography, the distribution of the soil moisture content was uneven, the soil moisture content thrown by the seed belt was high and a small number of broken straw sticks on both sides of the seed belt affected the sowing depth and stability of the corresponding seed belt. Overall, the simulation optimization results were in good agreement with the field experiment, which proved the scientificity and accuracy of the experiment.

**Table 6.** Factors and levels of experiment.

| With Species Serial Number | Test Number | Test Index | |
| --- | --- | --- | --- |
| | | Variation Coefficient of Sowing Depth/% | Coefficient of Variation of Lateral Uniformity/% |
| 1 | 1 | 4.82 | 12.67 |
| | 2 | 5.08 | 12.38 |
| | 3 | 5.19 | 12.59 |
| | 4 | 4.63 | 12.44 |
| 2 | 1 | 4.56 | 13.19 |
| | 2 | 4.93 | 13.68 |
| | 3 | 4.71 | 12.94 |
| | 4 | 4.24 | 14.11 |
| 3 | 1 | 4.19 | 12.21 |
| | 2 | 3.56 | 11.57 |
| | 3 | 4.20 | 12.39 |
| | 4 | 4.73 | 12.59 |

## 4. Discussion

At present, wheat wide-seed-belt sowing is becoming a hot research topic in China, and the operation performance of this model directly affects or even determines the success of this model. By building a discrete element simulation platform for the model of the soil-throwing device after wheat was sown using a wide seed belt, the rotation speed of rotary blade shaft, the angle of the soil guide plate and the angle of the soil retaining plate were taken as variables, and the variation coefficient of the wheat-sowing depth and lateral uniformity were taken as evaluation indexes. This paper analyzed the influence of three variables on the performance of throwing soil after a wide seed belt was used for sowing. In recent years, in order to solve the problem of unstable sowing depth and uneven sowing, Jiang et al. [21] carried out a field test on a wide seed belt sowing and fertilizing machine for wheat. The variation coefficient of sowing depth and uniformity was 4.15% and 11.55%, and the whole machine design meets the agronomic requirements of field sowing. Aiming at the problem that the poor stability of sowing depth affects sowing quality, Cheng et al. [22] designed a sowing device with equal depth and a wide seeding belt for wheat by means of mechanical actuator optimization and electrical control. The experimental results showed that the variation coefficient of the sowing depth was 6.8% when the theoretical sowing depth was 0.03 m. Zheng et al. [3] designed a soil shunt type wheat seeder with wide seedling belt and less tillage, the variation coefficients of sowing depth were 27.75% during a simulation test. Zhu et al. [23] set up a discrete element simulation platform by taking the elastic seed plate of a wide seedling belt seeder as the research object. The average variation coefficient of wheat grain lateral uniformity in a simulation experiment and field experiment was 12.10% and 13.40%, respectively.

Comparing the experimental results with those of these experts, it was found that the variation coefficient of sowing depth is similar to that of Jiang et al. [21] and 2.4% lower than that of Cheng et al. [22], this difference indicates that the sowing depth of this wide seed belt is consistent. Meanwhile, our results are 23.4% lower than those of Zheng et al. [3]. As for the variation coefficient of sowing uniformity, the variation coefficient of the wheat grain lateral uniformity between the simulation test and the field test is 0.36% and 0.67% different from the results of Zhu et al. [23], which shows a high similarity and also proves the scientific nature of the test to some extent. At the same time, it is 2.4% lower than the results of Cheng et al. [22] and Zheng et al. [3].

Starting from the effect of the soil covering the wheat after sowing, this study took the variation coefficient of sowing depth and the variation coefficient of sowing lateral uniformity as indicators and made some innovations in the structure design, working principle and test method of the test bed, but there were some limitations, as follows:

(1) Due to subjective and objective factors, such as time and conditions, a small amount of broken straw and fertilizer contained in the soil was not considered in the simulation process. The next step is to improve the simulation environment and conditions and to comprehensively and deeply study the movement characteristics and laws of wheat seeds under various particle conditions.

(2) Limited by the research scheme and test conditions, the test object in this paper was only one kind of soil environment, but the water content, viscosity and friction factors of different soil types are different, so more soil types will be studied in the follow-up test.

(3) After the wheat seeds were sown and had landed, the soil particle covering impacts and disturbs the wheat. Through high-speed photography technology and the gas–solid coupling method of the 6-DOF dynamic grid model, the interaction law between the soil and the wheat seeds and the characteristics of the movement track of the wheat seeds in the throwing and covering device after the wheat seeds are sown using the wide seed belt can be further explored.

(4) The test bed developed in this paper meets the requirements of existing wide-seed-belt sowing and achieves the goal of reducing manpower investments and improving working efficiency. According to the expected plan, it should be compared with the traditional, conventional mechanized sowing method (supporting straw crushing and

returning machine, rotary tiller and wheat seeder). However, because of the change of plan due to the weather, in the next step, a controlled experiment can be conducted in the same environment.

## 5. Conclusions

Based on the agronomic characteristics of the rice–wheat rotation area in the middle and lower reaches of the Yangtze River, this paper designed a test bed for sowing and covering the soil on a wheat wide seed belt. Once in the ground, it can complete the functions of spreading the wheat in stripes after straw crushing, spreading the wheat evenly with a wide seed belt and covering the soil evenly after sowing. It scientifically demonstrated that the wide seed belt on both sides of the straw strips, and the soil was rotated and then thrown across the seed guide tube, the seeds were evenly sown on the soft seed bed after rotary tillage and the flying soil was rotated and covered the seeds, innovatively. The main conclusions are as follows:

(1) The structural characteristics and technological principle of the test bed for sowing and covering the soil using a wide seed belt were in line with the agronomic requirements for sowing wheat with a wide seed belt and the technical characteristics needed for planting wheat with less tillage in the middle and lower reaches of the Yangtze River, which was conducive to improving the uniformity of sowing depth and lateral uniformity of sowing wheat with wide seed belt and to the later growth of wheat.

(2) Taking the variation coefficient of the sowing depth and the variation coefficient of sowing lateral uniformity as indexes, the results of single factor experiments showed that the rotating speed of the rotary blade shaft was 260–300 rpm, the angle of soil guide plate was 36°–48° and the angle of soil retaining plate was 58°–74°. Through the Box–Behnken simulation test, the optimal parameter combination is as follows: rotary blade shaft speed 282.1 rpm, soil guide plate angle 42.4° and soil retaining plate angle 65.5°. At this time, the variation coefficients of sowing depth and the sowing lateral uniformity of the simulation test were 4.35% and 12.46%, respectively.

(3) The variation coefficient of sowing depth and the variation coefficient of sowing lateral uniformity were 4.57% and 12.73%, respectively, under the field verification test. The simulation results of the simulation test and field test were basically consistent with the field test, which proved the rationality and accuracy of the test. The test results met the requirements of the no-tillage sowing of wheat with a wide seedling belt in the middle and lower reaches of the Yangtze River. The area covered by this seeding method will be further increased in the future, and it will also bring large-scale seeding quality.

**Author Contributions:** Conceptualization, B.W. and F.G.; methodology, Z.H. and W.L.; software, B.W. and X.C.; validation, B.W. and W.L.; formal analysis, F.G.; investigation, F.W.; resources, F.G.; data curation, B.W.; writing—original draft preparation, B.W.; writing—review and editing, B.W.; visualization, F.G. and X.C.; supervision, F.G.; project administration, B.W.; funding acquisition, B.W. All authors have read and agreed to the published version of the manuscript.

**Funding:** This research was funded by the following fund projects: 1. The Natural Science Foundation of Jiangsu Province, grant number BK20221187; 2. National Peanut Industry Technology System, grant number CARS-13.

**Institutional Review Board Statement:** Not applicable.

**Informed Consent Statement:** Not applicable.

**Data Availability Statement:** The datasets used and/or analyzed during the current study are available from the corresponding author upon reasonable request.

**Acknowledgments:** The authors would like to thank their teacher and supervisor for the advice and help during the experiments. We also appreciate the editor and anonymous reviewers for their valuable suggestions for improving this paper.

**Conflicts of Interest:** The authors declare no conflict of interest.

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
