# Peer review of "Analysis and Evaluation of Influencing Factors on Uniform Sowing of Wheat with Wide Seed Belt after Sowing and Soil Throwing Device"

_agriculture, doi:10.3390/agriculture12091455_

Round 1
Reviewer 1 Report
In this study, a test-bed for throwing soil after sowing with wide wheat seed belt was designed, which could complete the functions of straw crushing, straw lateral concentration and uniform sowing at one time to solve the problem of uneven sowing of wheat on the ground covered with rice straw in the rice-wheat rotation area in the middle and lower reaches of the Yangtze River. The study is very interesting in respect with sowing methods. The results are explained both by experiments and simulations. The experimental results satisfy with simulation results well. The author handle the manuscript well. The manuscript in its present form is suitable for publication after some queries that are given in the attached file.

Reviewer 2 Report
The manuscript entitled “Analysis and evaluation of influencing factors on uniform sowing of wheat with wide seed belt after sowing and soil throwing device" which focuses the problem of uneven sowing of wheat on the ground covered with rice straw. The paper is well written and structured, but the paper lacks key scientific insights. Minor shortcomings with the manuscript are as follows;
· Abstract needs to be improved on the qualitative basis (only necessary information needs to be added).
· Extensive literature review is missing in introduction.
· The overall graphics of the paper are good, but the text font written inside the Fig’s must be same with the rest font of the paper. Some Figures gets blur when they zoom in.
· Scientific writeup should be improved throughout the manuscript.
· In the methodology section, kindly revise the section 2.4.1 and 3.2.2 be more specific.
· Make the Figure 2, Figure 3, and Figure 4 standalone
· For the case of equations, try to give more investigative parameters that conceptualize and enhance the model understanding. Also take cautious about the use of super-scripts, sub-scripts, capitalization, and symbols.
· It could be better and make things clear to explain the results and discuss them in one heading. The separate heading for the discussion could be merged with results heading.
· The results obtained can be useful for future research as well if the variables being researched are well-defined and their nature i.e., dependent, independent etc. is clarified in the methods section. Furthermore, how this study could be useful in future.
· How about adding the cost analysis and the tractor speed on the sowing for further strengthening the study.
· Revise the conclusion. Make it concise and only necessary information must be discussed, avoid the irrelevant and repetition of the text.
Reviewer 3 Report
Substantive assessment:
Strip cereal sowing has been used successfully in Europe and the USA for many years (e.g. the German company HORSCH produces such seeders, and in Poland, Czajkowski and MZURI). Good for large acreages cultivated in monocultures and on good soils, e.g. Ukrainian loess. In addition to the works of Chinese researchers (many self-citations), it is also worth including in the bibliography valuable studies from other continents, where the agricultural culture is also at a high level, e.g. in North America (the strip-till method was developed in the USA) and the aforementioned Europe. The described method also has disadvantages, which farmers notice, and still use the classic methods of sowing cereals and soil preparation on lower-class soils. In this case, there is a specific forecrop, i.e. rice, so this research is utilitarian only for this region. The authors presented the problem very clearly and solve it with the use of modern computer and statistical methods and techniques.
Very good visualization, engineering drawings and allow you to quickly understand the essence of the problem. You can improve the proportions in Figure 1, where the drilling depth (3 cm) is almost equal to the width of the stripes (25 cm).
It is also worth mentioning the durability and reliability tests of agricultural machines, this also applies to prototypes. These machines, which are expensive to buy, in addition to functionality, must also be reliable in operation, which is what farmers require. Examples of reliability quantification methods have also been recently published in the MDPI Publishing House: https://doi.org/10.3390/agronomy12061364https://doi.org/10.3390/ma14227014
MDPI publications are worth and should be cited, because the authors benefit from it (higher IF).
Conclusions 2 and 3 are only a repetition of previous results. They should be more general and indicate the perceived advantages, but also disadvantages of such a sowing technique and plans for the future. There is also no economic account, and today a farm is a self-financing enterprise. The strip sowing method may be very good, but will they be willing to use it?
Editorial Rating:
- giving the speed in "r/min" - better and according to the SI it is "rpm"
- additionally, the RPM value should rather be an integer, e.g. 282 rpm (line 29 Abstracts)
- to define a range of values, for example angles or rotations, it is better to use a hyphen (-), because the tilde symbol (~) means an approximation, eg around ~230 V. Tilde has many other meanings apart from mathematics. In the SML and Oz programming languages, ~ means the negation of a number, for example, ~4 means -4 (minus 4).
- a lot of wrong entries in other units (e.g. lines 216, 219, 258, 291 ...)
- instead of a value that is difficult to understand and imagine, e.g. 0.0008~0.0012 m, it is better to get rid of these zeros to 0.8-1.2 mm
- formulas 4 and 5 do not need to be specified, because it is obvious
- one method of accuracy of a given parameter should be used throughout the work (the same number of decimal places), e.g. in line 299.
